# Fatty Acid β-Oxidation May Be Associated with the Erythropoietin Resistance Index in Stable Patients Undergoing Haemodialysis

**DOI:** 10.3390/diagnostics14202295

**Published:** 2024-10-16

**Authors:** Shuhei Kidoguchi, Kunio Torii, Toshiharu Okada, Tomoko Yamano, Nanami Iwamura, Kyoko Miyagi, Tadashi Toyama, Masayuki Iwano, Ryoichi Miyazaki, Yosuke Shigematsu, Hideki Kimura

**Affiliations:** 1Department of Clinical Laboratory, University of Fukui Hospital, Fukui 910-1193, Japan; skido@u-fukui.ac.jp (S.K.); k-torii@fukui-med.jrc.or.jp (K.T.); okadat@u-fukui.ac.jp (T.O.); tyama@u-fukui.ac.jp (T.Y.); inui@u-fukui.ac.jp (N.I.); ttoyama@u-fukui.ac.jp (T.T.); 2Department of Clinical Laboratory, Japanese Red Cross Fukui Hospital, Fukui 918-8501, Japan; 3Department of Internal Medicine, Fujita Memorial Hospital, Fukui 910-0004, Japan; miyagi@fujita-mhp.jp (K.M.); ryoichi@mitene.or.jp (R.M.); 4Division of Nephrology, Department of General Medicine, School of Medicine, University of Fukui, Fukui 910-1193, Japan; 5Department of Pediatrics, Faculty of Medical Sciences, University of Fukui, Fukui 910-1193, Japan; yosuke3321181@gmail.com

**Keywords:** renal anaemia, erythropoietin resistance index, haemodialysis, fatty acid β-oxidation, acylcarnitine, reticulocyte, mitochondria, adiponectin, lipid profile

## Abstract

Background/Objectives: Lipid metabolism and adiponectin modulate erythropoiesis in vitro and in general population studies and may also affect responsiveness to erythropoietin in patients undergoing haemodialysis (HD). However, little is known about the impact of lipid-associated biomarkers on reticulocyte production and erythropoietin resistance index (ERI) in patients undergoing HD. Therefore, we aimed to investigate their impacts in 167 stable patients undergoing HD. Methods: Pre-dialysis blood samples were collected and analysed for reticulocyte counts and serum lipid profiles by routine analyses and serum carnitine profiles (C0–C18) by LC-MS/MS. ERI was calculated as erythropoietin dose/kg/week normalized for haemoglobin levels. Results: The independent positive determinants of reticulocyte count were log [Triglyceride (TG)] and logC18:1. A large proportion of longer-chain acylcarnitines was positively correlated with reticulocyte counts, possibly resulting from the accumulation of acylcarnitines in mitochondria undergoing fateful exocytosis from reticulocytes. These results indicate a possible association between reticulocyte formation and reduced β-oxidation, which occurs during the peripheral phase of erythroblast enucleation. Total cholesterol (TC) and log [C2/(C16 + C18:1)] as a putative marker of β-oxidation efficiency were negative independent determinants of ERI. Moreover, acyl chain length had a significantly positive impact on the correlation coefficients of individual acylcarnitines with ERI, suggesting that enhanced β-oxidation may be associated with reduced ERI. Finally, adiponectin had no independent association with reticulocyte counts or ERI despite its negative association with HDL-C levels. Conclusions: Enhanced fatty acid β-oxidation and higher TC levels may be associated with lower ERI, whereas higher TG levels and longer acylcarnitines may be related to the latest production of reticulocytes in stable patients undergoing HD.

## 1. Introduction

Anaemia of chronic kidney disease (CKD) is a common complication in patients undergoing haemodialysis (HD), mainly due to the reduced renal production of erythropoietin, and is associated with a reduced quality of life and poor prognosis in this setting [1,2]. Although several types of recombinant human erythropoietin are widely used to treat anaemia, the amount required to maintain target haemoglobin levels varies among patients [3]. Furthermore, a non-negligible number of patients undergoing HD exhibit hyporesponsiveness to erythropoietin despite its high doses [1,3,4]. As higher doses of erythropoietin are associated with poor prognosis [1,3,4], clinical factors affecting the erythropoietin resistance index (ERI) have been investigated for more effective use of erythropoietin [5]. Iron deficiency, inflammation, uraemia, bone metabolic factors, and non-iron malnutrition have been reported as ERI-enhancing factors [5]. 

Notable associations between lipid markers and erythropoiesis have recently been reported in the general population with normal to mild renal dysfunction [6,7,8]. Serum levels of adiponectin, an anti-atherosclerotic adipocytokine, increase in patients with decreased renal function and are negatively correlated with the current values of haemoglobin (Hb), in addition to being an independent risk factor for the subsequent development of anaemia [6,7]. In the general population, triglyceride (TG) is an independent positive determinant of red blood cell (RBC) count, whereas adiponectin is a negative determinant [8]. Adiponectin has lipid-modulating effects [9,10] and an inhibitory effect on erythropoiesis; its higher values may reflect greater adipogenesis in the bone marrow [6,8]. 

Considering that carnitine and acylcarnitine are involved in the lifespan of erythrocytes [11,12], their influence on erythropoietin dose or ERI in patients undergoing dialysis has been investigated [13,14,15]. These analyses showed that carnitine deficiency and the accumulation of medium-to-long-chain acylcarnitines were associated with higher ERI [13,15] and erythropoietin doses [14], suggesting that fatty acid oxidation may be associated with ERI. Notably, the division and proliferation of haematopoietic stem cells and erythroblasts requires an energy shift from glycolysis to fatty acid β-oxidation in the mitochondria [16,17,18,19]. Moreover, erythroblast enucleation reportedly inhibits the TCA cycle [20], which may suppress fatty acid oxidation. Thus, erythropoiesis may be closely linked to β-oxidation status. However, no study has explored the association of these lipid markers with reticulocyte count, a proximal effect of erythropoietin use, in patients undergoing HD. No studies have analysed the association of acylcarnitine profiles with ERI from the perspective of β-oxidation. Furthermore, the actual effects of L-carnitine as an adjunctive therapeutic agent for anaemia of CKD vary among studies, and the underlying reasons remain unclear [12,21].

We measured the serum levels of adiponectin, routine lipid profiles, and free carnitine and acylcarnitine profiles in 167 patients undergoing HD to investigate the relationship between these lipid markers and erythropoietic status, indicated by reticulocyte count and ERI, to obtain novel evidence for the optimal management of anaemia of CKD. 

## 2. Materials and Methods

### 2.1. Patients and Study Design

This observational study was conducted in April 2014 at the Fujita Memorial Hospital and the Department of Clinical Laboratory, University of Fukui Hospital. This study was conducted in accordance with the Declaration of Helsinki and was approved by the ethics committees of Fujita Memorial Hospital (approval code 24 and date of approval: 7 September 2012) and the University of Fukui Hospital (approval code 20120137 and date of approval: 2 November 2012). Written informed consent was obtained from all the participants. 

For this study, we recruited 167 patients undergoing maintenance HD treatment at Fujita Memorial Hospital. The inclusion criteria were as follows: (1) aged 18 years or more, (2) thrice-weekly 3 h to 5 h HD sessions using high-flux membranes and standard heparin doses, (3) HD vintage over 3 months, and (4) C-reactive protein (CRP) less than 0.3 mg/dL [22] and ferritin less than 300 ng/mL [23]. The exclusion criteria were as follows: (1) no treatment with recombinant human erythropoietin, (2) treatment with L-carnitine, (3) active infectious diseases, (4) active liver disease or cancer, (5) recent blood transfusion or surgical procedure, and (6) active haemorrhage. HD was initiated for end-stage renal disease owing to chronic glomerulonephritis (*n* = 64), diabetic nephropathy (*n* = 47), nephrosclerosis (*n* = 15), polycystic kidney disease (*n* = 14), and other conditions (*n* = 27). Smoking was defined as current or habitual cigarette smoking. Hypertension was defined as systolic blood pressure > 140 mmHg, diastolic pressure > 90 mmHg, or the use of antihypertensive drugs. The presence of diabetes mellitus (DM) was determined based on current medication or medical history. 

### 2.2. Definition of ERI

The three types of erythropoiesis-stimulating agents (ESA) used were epoetin β (EPOβ, once to three times weekly, *n* = 7), darbepoetin α (DA, once weekly, *n* = 79), and epoetin β pegol (EPOP, once every 4 weeks, *n* = 81). Units of EPOβ were converted to micrograms using a ratio of 200:1 [4]. Considering the 96% occupancy of DA and EPOP and an EPOP use interval of 4 weeks, the ERI was defined as the 4-week total dose of ESA (over the previous 4 weeks) (μg)/dry weight (kg) /Hb (g/dL). All patients were treated in accordance with the guidelines for anaemia of CKD [1,24]. According to the guidelines, resistance to ESA was defined as failure to achieve the target Hb level of 10 g/dL or more despite the use of 9000 IU/week (EPOβ), 60 μg/week (DA), or 240 or more μg/week (EPOP) [23,24]. 

### 2.3. Biochemical Assays

Venous blood samples (serum and whole blood) were collected immediately prior to the first HD, 2 h after breakfast, or 2–3 h after lunch, except for post- and next pre-dialysis urea nitrogen for Kt/V (urea kinetic measure) and protein catabolic rate (PCR) from all dialysis patients. Blood collection was performed 7 days after the injection of DA (*n* = 79) or EPOP (*n* = 81), or 3 days after EPOβ injection (*n* = 7). Laboratory data included routine blood tests, blood chemistry, serum iron (Fe), serum ferritin, total iron-binding capacity (TIBC), serum zinc, C-reactive protein (CRP), and intact parathyroid hormone (iPTH) levels measured using automated and standardised methods. Kt/V and PCR were performed using a single-pool urea kinetic model. Transferrin saturation (TSAT) was determined using Fe and TIBC measurements. Total adiponectin levels were measured using a newly developed automated homogenous assay (Denka Seiken Co., Ltd., Tokyo, Japan) on a TBA-c16000 chemistry analyser (Canon Medical Systems Corp., Tochigi, Japan). 

### 2.4. Mass Spectrometry Measurements

Serum samples were separated immediately after blood collection in a refrigerated centrifuge and stored at −80 °C until analysis. Serum (5 μL) was mixed with 200 μL of a methanol solution of stable isotope-labelled free carnitine and acylcarnitines as the internal standard. The constituents were mixed by vortexing for 1 min followed by centrifugation at 16,000 rpm for 10 min at 4 °C. Flow injection and electrospray ionisation tandem mass spectrometry (MS/MS) analyses were performed using the API 4000 LC/MS/MS system (AB Sciex, Tokyo, Japan). A 5 µL sample was introduced into the liquid chromatography flow of acetonitrile/water (4:1) with 0.05% formic acid. Positive-ion MS/MS analysis was performed in precursor ion scan mode with an 85 m/z product ion. The coefficients of variation for the quality control samples, including free carnitine and each acylcarnitine, were between 2.8% and 9.3%. As for free carnitine, carnitine deficiency, insufficiency, and sufficiency were defined as serum concentrations of less than 20 nmol/mL, 20 to less than 36 nmol/mL, and 36 to 74 nmol/mL, respectively [21].

### 2.5. Statistical Analysis

The Kolmogorov–Smirnov test was used to determine the normality of the measurement distribution. Normally distributed continuous variables are expressed as mean ± standard deviation (SD) and non-normally distributed values as medians and interquartile ranges (IQR). Categorical variables were described as frequencies and percentages. Non-normally distributed data were log_10_ transformed to reduce pronounced positive skews and decrease variation. The differences in continuous variables between the two groups and among the three groups were assessed using unpaired t-test and ANOVA with post hoc multiple comparisons, respectively. Chi-square analysis was used to assess the differences in discontinuous variables between the groups. Univariate and multivariate linear regression analyses were performed to investigate the factors associated with reticulocyte count and ERI. Multivariate linear regression analyses using forward stepwise or forced entry methods were performed using a model that included clinical parameters significantly correlated with reticulocyte counts or ERI. Partial correlation coefficients were obtained after adjusting for age, sex, and HD vintage to investigate the associations between individual acylcarnitine levels, reticulocyte counts, and ERI. The resulting correlation coefficients were then categorised by the number of carbon atoms contained in each acylcarnitine acyl chain, and linear regression models were generated to determine the impact of acyl chain length on the relationship between acylcarnitines and reticulocyte counts and ERI. The standard bootstrap method with 2000 bootstrap samples with a 95% confidence interval (CI) was used to confirm the stability of our study model. 

Statistical significance was set at *p* < 0.05 (two-tailed tests). All statistical analyses were performed using the SPSS statistical software package ver. 24.0 (SPSS Inc., Chicago, IL, USA) and R software, version 4.3.1. 

## 3. Results

### 3.1. Baseline Characteristics

A total of 167 patients undergoing HD aged 66.9 ± 12.3 years (102 men) were enrolled in this study. Table 1 presents the detailed clinical characteristics of all the patients. None of the patients had any inherent errors in their organic or fatty acid metabolism. The mean TSAT was 23.3%, and the mean Fe level was 61.0 ng/mL. 

### 3.2. Correlation of Carnitine Profile with Reticulocyte Counts and ERI

Twenty-two carnitine species were also assessed. Very-long-chain acylcarnitines with >18 carbons were not measured. The carnitine profile and its partial correlation with the reticulocyte count and ERI are shown in Table 2. The partial correlation coefficients were determined after controlling for age, sex, and HD vintage. A wide range of acylcarnitines was significantly positively correlated with reticulocyte counts, with longer-chain acylcarnitines having greater coefficient values. None of the acylcarnitines were significantly correlated with the ERI, except for C8:1, whereas log [C2/(C16 + C18:1)], a putative marker of β-oxidation efficiency [25], was significantly negatively associated with the ERI. The C2/(C16 + C18:1) value is an indicator of better fatty acid β-oxidation and is negatively associated with all-cause mortality in patients undergoing HD [25], while the reciprocal value is an indicator of poor β-oxidation and is used as a screening test for children with β-oxidation disorders [26].

### 3.3. Determinants of Free Carnitine and Characteristics According to Carnitine Levels

In the entire group, age, HD vintage, and Kt/V were independent negative determinants of serum carnitine levels, whereas PCR was an independent positive determinant (Appendix A). Overall, serum levels of free carnitine were less than 20 nmol/mL in 48 (Group 1, G1; 28.7%) patients, satisfying the definition of carnitine deficiency, whereas 99 (Group 2, G2; 59.3%) patients had carnitine levels below the normal lower limit (36 nmol/mL), designated as carnitine insufficiency. The remaining 20 patients (Group 3, G3; 12.0%) had carnitine levels within the reference range (36–74 nmol/mL).The characteristics of G1-3 are shown in Table 3. The carnitine deficiency group (G1) had significantly higher age, Kt/V, and adiponectin levels and lower zinc levels, dry weight (DW), and PCR than the G2 and G3 groups. These factors may account for the malnutrition and lower carnitine levels in group G1. Remarkably, G1 also demonstrated a significantly higher ERI and lower log [C2/(C16 + +C18:1)], a putative marker of β-oxidation efficiency [25], compared to G2 or G3.

### 3.4. Independent Determinants of Reticulocyte Counts and ERI

Table 4 and Table 5 shows the clinical determinants of reticulocyte counts in univariate and stepwise multivariate linear regression analyses, respectively. The positive independent determinants were blood levels of log TG, log C18:1, and platelets (Plt), whereas the negative independent determinants were serum Fe levels and male sex (Table 5). In G1, logC18:1 was a positive independent determinant (Appendix A). As for ERI, univariate (Table 6) and stepwise multivariate (Table 7) linear regression analyses reveal that total cholesterol (TC), log[C2/(C16 + C18:1)], Fe, Plt, and the presence of DM were negative independent determinants of ERI. Adiponectin was not associated with reticulocyte counts and had a univariate, but not multivariate, association with ERI. 

### 3.5. β-Oxidation Promotion May Lead to Reduced ERI

As shown in Table 2, shorter-chain acylcarnitines (C2-C8) were negatively associated with ERI, whereas longer-chain acylcarnitines (C14-C18), except for C16OH and C18, were positively associated with ERI. Although the associations were not statistically significant, they indicated that acyl chain length had a positive impact on the correlation between acylcarnitines and ERI, implying that β-oxidation promotion may be associated with reduced ERI. To verify this hypothesis, linear regression analysis was performed between the length of the acylcarnitine acyl chains and the partial correlation coefficients between each acylcarnitine and ERI. Figure 1 and Appendix A show the corresponding scatter plots with trend lines, indicating the results of the regression analysis. The direction of the significant association revealed that the shorter acyl chains of acylcarnitines had a greater negative association with ERI in the whole group, as well as in G1, G2, and G3. 

### 3.6. Lipid Markers in Patients Who Are Resistant or Respond to ESA

Based on the definition of ESA resistance described in the Patients and Methods section, five patients were identified as having ESA resistance (full range of ERI: 0.419–0.719), and the remaining patients (*n* = 162) were considered ESA-responsive (full range of ERI: 0.040–0.575). The ESA-resistant patients had a significantly lower level of TC (129.0 ±2 6.3 vs. 159.4 ± 26.3, *p* < 0.05) than the ESA-responsive patients, while the two groups did not differ in log[C2/(C16 + C18:1)] values. When the lowest ERI value of ESA-resistant patients (0.419) was used as a cutoff for putative ESA resistance, the sensitivity and specificity were 100% and 95%, respectively, with positive and negative predictive values of 38% and 100%. Patients with a higher ERI (0.419 or higher, *n* = 13) had a significantly lower TC (135.8 ± 31.5 vs. 160.4 ± 30.8, *p* < 0.01) and a trend towards lower values of log[C2/(C16 + C18:1)] (1.292 ± 0.143 vs. 1.366 ± 0.144, *p* < 0.08) than those with a lower ERI (<0.419, *n* = 154). Furthermore, when the median ERI (0.196) was used as a cut-off for putative ESA resistance, patients with higher ERI (*n* = 85) tended to have lower TC levels (154.0 ± 31.2 vs. 163.1 ± 31.3, *p* < 0.07) and had significantly lower log[C2/(C16 + C18:1)] values (1.333 ± 0.147 vs. 1.388 ± 0.138, *p* < 0.05) than those with lower ERI (*n* = 82). 

## 4. Discussion

Lipid metabolisms modulate erythropoiesis in vitro and in general population studies [8,16,19,20] and may also affect responsiveness to erythropoietin in patients undergoing HD. However, more research needed to understand their impacts on erythropoiesis status in this setting, and we therefore attempted to clarify these issues in the present clinical study. In our entire HD cohort, the positive independent determinants of reticulocyte counts were log TG, log C18:1, and Plt, whereas the negative independent determinants were serum Fe levels and male sex. In the carnitine-deficient group (G1), logC18:1 was a positive independent predictor. A large proportion of short- to long-chain acylcarnitines was positively correlated with the reticulocyte count. Furthermore, the blood levels of TC, log[C2/(C16 + C18:1)] as a putative marker of β-oxidation [25,26], Fe, Plt, and the presence of DM were negative independent determinants of ERI in the entire group. G1 demonstrated a lower log [C2/(C16 + C18:1)] value and higher ERI than the carnitine-insufficient groups (G2), the other main group. Moreover, in all groups, acyl chain length had a significant impact on the association between acylcarnitines and ERI: short-chain acylcarnitines were negatively correlated with ERI, whereas long-chain acylcarnitines were positively associated with ERI. Our findings demonstrate that TG, TC, and acylcarnitine profiles are related to erythropoiesis status in patients undergoing HD, revealing clinically for the first time that enhanced β-oxidation may be independently associated with reduced ERI and extending possible associations of multiple lipid metabolism with erythropoiesis. 

Recent studies have reported that adiponectin is negatively associated with erythropoiesis in individuals with normal to mild renal dysfunction, probably owing to the suppression of erythropoiesis in the bone marrow [6,7,8]. We found a univariate, but not multivariate, positive association between adiponectin and ERI in the entire group. Other confounding factors for ERI, such as DW, may have diminished the erythropoiesis-inhibiting effect of adiponectin. However, adiponectin elevation induced by malnutrition and reduced renal clearance also increases high-density lipoprotein cholesterol (HDL-C) levels and reduces TG levels, owing to its essential function in modulating lipids. 

One of the important findings of the current study was the first clinical report that logTG and logC18:1 were independently correlated with reticulocyte count, which supported the results of earlier experimental studies on erythropoiesis and mitochondrial β-oxidation, described below. As the counts were measured approximately 7 days after erythropoietin administration, they presumably reflected a near-peak response to erythropoietin [27,28], whereas erythropoietin-stimulated proliferation from burst-forming unit-erythroid cells to reticulocytes takes approximately 7–9 days [29]. Notably, the proliferation and differentiation of progenitor cells into erythroblasts requires an energy shift from glycolysis to fatty acid β-oxidation in the mitochondria [16,17,18,19]. Higher serum TG levels may supply more fatty acids to the bone marrow and promote erythroid cell proliferation because erythropoietin also induces plasma lipoprotein lipase in patients undergoing HD [30]. The existence of long-chain acylcarnitines, such as C18:1, in the blood can be explained by the following mechanisms. Medium- (C8-12) and long (C14-18)-chain acylcarnitines are formed from acyl CoA and carnitine via carnitine acyl CoA transferase (CACT) on the mitochondrial outer membrane and then enter the mitochondrial matrix via carnitine acyl-carnitine translocase. They are then reconverted to acyl CoA by CACT in the matrix and donated for β-oxidation, which begins with longer acyl chains that are shortened to produce acetyl CoA as a source of energy [31]. At equilibrium, medium- to long-chain acyl-CoA can be converted to acylcarnitine to maintain free CoA levels, and an appropriate amount of the resultant acylcarnitines may be transferred to the cytosol via retrograde pathways. Therefore, compared with short (C2-6) chain acylcarnitines, which can freely move between the cytoplasm and mitochondria, longer chain acylcarnitines diffuse less freely toward the cytosol or outside cells and are generally confined to the mitochondria [21,31]. Just before erythroblast enucleation, the ATP production pathway shifts from β-oxidation to glycolysis [20], probably leading to the suppression of β-oxidation and the accumulation of long-chain acylcarnitines in the mitochondria. Finally, mitochondria undergo exocytosis from reticulocytes [32,33,34,35], whereby mitochondrial acylcarnitines are released into the bloodstream. Consequently, acylcarnitines in exocytosed mitochondria can be positively and partially correlated with reticulocyte counts, which explains the positive correlation between reticulocyte counts and long-chain acylcarnitines, mainly C18:1, observed in the current study. However, considering the at least several-fold-higher amounts of acylcarnitines in the RBC than in the serum [36] and the shorter survival time of RBC in patients undergoing HD as reported earlier [37], blood acylcarnitine levels may also mirror traces of the β-oxidation status during the long process of proliferation and maturation of whole erythroid cells, with greater cell numbers overwhelming the other cells. 

In this study, we identified male sex and Fe levels as independent negative determinants of reticulocyte counts, which was contrary to our general expectations. No significant differences between men and women were found in age, HD vintage, DM presence, log TG, amounts and types of ESA used, logCRP, Fe, Plt, WBC, or intravenous iron supplementation. In contrast, men showed a trend toward lower levels of alkali phosphatase (ALP: 228 ± 75.3 U/L vs. 255.6 ± 102.1 U/L, *p* < 0.06) than women. Furthermore, ALP levels had a univariate positive association (β = 0.180, *p* = 0.020) with reticulocyte counts (Table 4 and Table 5). These results may have led to the lower reticulocyte counts of male sex compared to female sex, even on stepwise multivariate analysis, although why ALP levels were positively associated with reticulocyte counts remains unclear. Regarding the relationship between Fe levels and reticulocyte counts, Fe levels were negatively associated with reticulocyte counts (r = −0.271, *p* < 0.05) in patients on EPOP (*n* = 81) who received one high dose (50–250 μg) but not those on DA (*n* = 79) or EPOβ (*n* = 7) who received one low dose (7.5–60 μg). A similar result was previously reported in which a high dose of EPOP treatment (100–500 μg) produced not only a significantly greater increase in reticulocyte counts but also a significantly greater decrease in TAST levels at 3 to 14 days than EPOβ treatment (15–100 μg) [38]. One high dose may increase reticulocyte counts and decrease Fe levels simultaneously via enhancing iron utility abruptly. Since ESA amounts were not included in the stepwise analysis model (Table 5), EPOP treatment may have caused the independent negative association between Fe levels and reticulocyte counts observed in our study. 

Another important finding was the negative correlation between log [C2/(C16 + C18:1)] and ERI and the positive impact of acyl chain length on the correlation between acylcarnitines and ERI in both the whole group and the lower carnitine groups (G1 and G2). Alternatively, short-chain acylcarnitines were negatively correlated with ERI, whereas long-chain acylcarnitines were positively correlated. Short-chain acylcarnitines, including acetylcarnitine (C2), are produced substantially from the promotion of β-oxidation, whereas long-chain acylcarnitines, mainly composed of C16 to C18, suggest the early failure of β-oxidation. Short-chain acyl-CoA, a precursor of short-chain acylcarnitines, can also be generated from glucose and amino acids [31,39]. Nevertheless, short-chain acylcarnitines may reflect higher β-oxidation and ATP production. Therefore, the negative associations of log [C2/(C16 + C18:1)] and shorter acyl chains with ERI indicate that β-oxidation promotion is associated with a lower ERI. Previous studies have reported that shorter acyl chains correlate with lower erythropoietin doses [14], and a short-chain (C5OH) or long-chain (C18) acylcarnitine had a negative or positive association with ERI [15], respectively, which is partially similar to our findings. It has recently been reported that β-oxidation of fatty acids plays an essential role in erythroid proliferation [16,17,18,19]. Fatty acid oxidation is involved in the asymmetric division of haematopoietic stem and progenitor cells [16,17]. Furthermore, fatty acid β-oxidation is required for erythropoiesis from pre-erythroblasts to terminal erythroblasts [18,19]. Therefore, considering these related clinical and experimental reports, our clinical data demonstrated for the first time the possibility that fatty acid β-oxidation status may be related to erythropoiesis in patients undergoing HD and that promoting β-oxidation may be associated with reduced ERI. Intriguingly, a recent study reported that an inhibitor of sodium-glucose transport 2 (SGLT2i) reduced the risk of anaemia events or the need for anaemia treatment in CKD with diabetes during a median follow-up period of 2.6 years [40]. Considering that SGLT2i enhances fatty acid oxidation in several organs [41], enhanced fatty acid oxidation may be associated with enhanced erythropoiesis in diabetic CKD.

Serum TC levels also showed a negative correlation with ERI in patients undergoing HD. This may be because cholesterol is necessary for haematopoietic stem cell maintenance [42]; therefore, the pool of stem and progenitor cells may have been larger in patients with high serum TC levels. Unexpectedly, DM was negatively and independently associated with ERI. In our study, patients with DM had significantly lower HD vintage and Fe levels and significantly higher WBC and Plt levels than those without DM. No significant differences in CRP levels were found between the two groups. Considering that shorter HD vintage and higher WBC and Plt levels were negatively associated with ERI in our study, patients with DM may have had greater productive ability of the bone marrow than those without DM. The positive correlation of reticulocyte counts with white blood cell and Plt counts and the negative correlation between Plt and ERI may be because erythrocytes are derived from granulocyte-macrophage/erythrocyte-megakaryocyte progenitor cells [43]. 

In addition, patients with ESA resistance or a higher ERI (a cutoff of 0.419 for a resistance level or 0.196 for the median) had significantly lower levels of TC or log[C2/(C16 + c18:1)] than those with an ESA response or a lower ERI. These results also showed that lower TC levels and reduced β-oxidation were directly associated with ESA resistance and higher ERI levels.

This study had several limitations. As the number of study participants was insufficient, other important determinants may not have been identified. Because some independent variables were not normally distributed, the bootstrap method was used to obtain more precise statistics and to perform internal validation of our results. As shown in Table 3, the age, HD vintage, DW, and serum albumin levels were different among the G1-3 groups. These factors may also be confounding factors for elevated ERI in the carnitine-deficient group (G1) compared with the other groups (G2 and G3). Since about 75% of L-carnitine is attained from diet, the dietary regimen may affect the carnitine level in the study population and can be considered as a confounding factor [44]. The interpretation of fatty acid oxidation is ambiguous, considering that the origin of short-chain acylcarnitines remains unclear. Furthermore, in patients with renal failure in whom pyruvate dehydrogenase and carnitine-palmitoyl transferase activities are reduced [45,46], acetyl CoA production from glycolysis and β-oxidation may be relatively decreased, and amino acids tend to be catabolised. Therefore, short-chain carnitine may be affected by pathways other than β-oxidation. 

## 5. Conclusions

In conclusion, our study demonstrated a unique association between lipid profile and erythropoiesis status in stable patients undergoing HD. LogTG and log C18:1 were independent positive determinants of reticulocyte counts, while TC and log[C2/(C16 + C18:1)], as a marker of β-oxidation efficiency, were independently and negatively associated with ERI. Moreover, acyl chain length had a significant positive impact on the correlation coefficients of individual acylcarnitines with the ERI. These findings suggest that greater β-oxidation of fatty acids leads to a lower ERI, clinically supporting the in vitro finding that β-oxidation is essential for the division and proliferation of erythroid cells. However, our study did not provide precise information for selecting the appropriate patients undergoing HD for L-carnitine treatment. A large-scale prospective survey of patients undergoing HD on L-carnitine is required to determine which changes in acylcarnitine profiles are associated with ERI reduction and to predict treatment efficiency. 

## Figures and Tables

**Figure 1 diagnostics-14-02295-f001:**
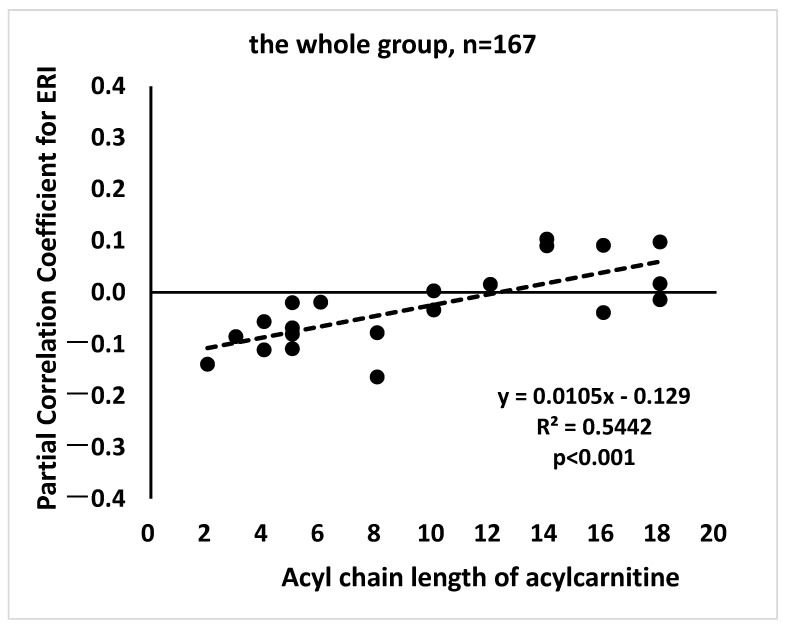
Scatter plots with a trend line between acylcarnitine chain length and partial correlation coefficients between each acylcarnitine and ERI show the results of the regression analysis. The partial correlation coefficients for ERI were adjusted for age, sex, and log (HD vintage).

**Table 1 diagnostics-14-02295-t001:** Characteristics of all the patients (*n* = 167).

Variables	Means ± SD or Medians [IQR] ^a^
Sex (male/female)	102/65
Age (year)	66.9 ± 12.3
Dry weight (kg)	55.5 ± 12.2
HD vintage (month)	81.0 [37.5, 169.0]
DM *n* (%)	47 (28.3%)
HT *n* (%)	141 (84.4%)
Smoking *n* (%)	23 (13.8%)
ACE inhibitor *n* (%)	18 (10.8%)
Albumin (g/dL)	3.7 ± 0.4
AST (U/L)	14.1 ± 5.5
ALT (U/L)	10.6 ± 5.6
ALP (U/L)	239.3 ± 87.4
LDH (U/L)	198.5 ± 46.1
CRP (mg/dL)	0.05 [0.03, 0.11]
WBC (/µL)	5233 ± 1591
RBC (10^4^/µL)	371.4 ± 35.4
Haemoglobin (g/dL)	11.0 ± 0.9
MCV (fL)	92.4 ± 6.2
Reticulocyte counts (10^4^/µL)	4.1 ± 1.9
PLT (104/µL)	14.9 ± 4.6
Fe (µg/dL)	61.0 ± 25.5
Ferritin (ng/mL)	26.8 [14.7, 49.4]
TIBC (µg/dL)	268.8 ± 45.3
TSAT (%)	23.3 ± 10.0
ESA (μg/4 weeks)	125.1 ± 59.6
ERI [μg(4 weeks)/kg/(g/dL)]	0.22 ± 0.12
iPTH (pg/mL)	131.1 ± 83.4
Zn (µg/dL)	64.4 ± 13.6
Kt/V	1.40 ± 0.28
PCR (g/day)	50.3 ± 13.8
Adiponectin (µg/mL)	21.7 ± 10.4
TC (mg/dL)	158.5 ± 31.4
HDL-C (mg/dL)	51.5 ± 14.1
LDL-C (mg/dL)	88.1 ± 26.5
TG (mg/dL)	81.0 [58.0, 113.0]

^a^ Normally distributed continuous variables were expressed as the means ± standard deviation (SD), and non-normally distributed values were expressed as medians and interquartile range (IQR).

**Table 2 diagnostics-14-02295-t002:** Free carnitine and acylcarnitine profiles and their partial correlation with reticulocyte counts and ERI in the whole group.

Variables ^a^	Means ± SD or Medians [IQR]	Partial Correlation ^a^			
		vs. Reticulocyte Counts		vs. ERI	
		r	*p* Value ^b^	r	*p* Value ^b^
C0 (free carnitine) (nmol/mL)	23.3 [19.0, 30.0]	0.115	0.139	−0.123	0.115
C2 (nmol/mL)	5.48 [4.19, 6.88]	0.206	**0.008**	−0.139	0.073
C3 (nmol/mL)	0.25 [0.19, 0.34]	0.197	**0.011**	−0.086	0.272
C4 (nmol/mL)	0.44 [0.31, 0.56]	0.120	0.124	−0.111	0.153
C4OH (nmol/mL)	0.05 [0.03, 0.06]	0.101	0.193	−0.057	0.468
C5 (nmol/mL)	0.16 [0.12, 0.21]	0.037	0.639	−0.069	0.380
C5:1 (nmol/mL)	0.05 [0.04, 0.07]	−0.095	0.223	−0.081	0.301
C5OH (nmol/mL)	0.10 [0.09, 0.13]	−0.127	0.103	−0.109	0.161
C5DC (nmol/mL)	0.85 [0.54, 1.09]	−0.053	0.496	−0.020	0.798
C6 (nmol/mL)	0.07 [0.04, 0.09]	0.148	0.056	−0.019	0.808
C8 (nmol/mL)	0.15 [0.11, 0.23]	0.141	0.070	−0.078	0.317
C8:1 (nmol/mL)	0.71 [0.46, 1.01]	0.174	**0.025**	−0.164	**0.035**
C10 (nmol/mL)	0.25 [0.19, 0.35]	0.128	0.100	0.003	0.969
C10:1 (nmol/mL)	0.18 [0.12, 0.24]	0.224	**0.004**	−0.034	0.664
C12 (nmol/mL)	0.10 [0.07, 0.13]	0.091	0.246	0.016	0.841
C14 (nmol/mL)	0.03 [0.02, 0.04]	0.281	**<0.001**	0.103	0.185
C14:1 (nmol/mL)	0.09 [0.07, 0.12]	0.175	**0.024**	0.090	0.247
C16 (nmol/mL)	0.10 [0.08, 0.12]	0.281	**<0.001**	0.091	0.242
C16OH (nmol/mL)	0.01 [0.01, 0.01]	0.075	0.340	−0.039	0.618
C18 (nmol/mL)	0.04 [0.03, 0.05]	0.250	**0.001**	−0.014	0.857
C18:1 (nmol/mL)	0.13 [0.11, 0.18]	0.343	**<0.001**	0.098	0.209
C18:1OH (nmol/mL)	0.01 [0.01, 0.01]	0.059	0.454	0.017	0.828
Non-C2AC/total carnitine	0.12 [0.10, 0.14]	0.032	0.686	0.057	0.464
C2/(C16 + C18:1)	23.3 [17.9, 29.0]	−0.080	0.306	−0.247	**0.001**
Total acyl carnitine (nmol/mL)	9.7 [7.4, 11.7]	0.215	**0.005**	−0.119	0.127
Total carnitine (nmol/mL)	33.6 [26.8, 42.1]	0.158	**0.041**	−0.124	0.110
Acyl carnitine/free carnitine	0.41 ± 0.10	0.158	**0.043**	0.040	0.606

^a^ After variables were log10 transformed, except for acyl carnitine/free carnitine, partial correlations were calculated adjusting for age, sex, and log(HD vintage). ^b^ *p*-values in bold are statistically significant.

**Table 3 diagnostics-14-02295-t003:** Characteristics of groups 1–3 classified according to free carnitine levels.

Characteristics	Group1 (*n* = 48) ^a^	Group2 (*n* = 99)	Group3 (*n* = 20)	*p*-Value ^b^
Sex (male/female)	25/23 ^d^	59/40 ^d^	18/2 ^c,e^	**0.013**
Age (year)	71.8 ± 10.1 ^c,d^	65.7 ± 12.0 ^e^	60.8 ± 14.8 ^e^	**0.001**
Dry weight (kg)	51.2 ± 10.0 ^d^	55.6 ± 10.9 ^d^	65.6 ± 16.8 ^c,e^	**<0.001**
HD vintage (month)	90.0 [43.5, 218.8] ^d^	84.0 [40.0, 169.0] ^d^	30.0 [6.8, 79.5] ^c,e^	**<0.001**
DM *n* (%)	17 (35.4%)	23 (23.2%)	7 (35.0%)	0.251
HT *n* (%)	39 (81.3%)	83 (83.8%)	19 (95.0%)	0.351
Smoking *n* (%)	7 (14.6%)	12 (12.1%)	4 (20.0%)	0.635
Albumin (g/dL)	3.6 ± 0.4	3.7 ± 0.3	3.8 ± 0.4	0.095
AST (U/L)	15.2 ± 5.2	13.6 ± 5.5	13.5 ± 6.1	0.226
ALT (U/L)	9.4 ± 4.5	10.8 ± 5.6	12.1 ± 7.6	0.156
ALP (U/L)	249.9 ± 106.7	231.4 ± 68.9	253.0 ± 115.4	0.370
LDH (U/L)	194.3 ± 36.7	199.7 ± 47.9	202.2 ± 57.6	0.741
CRP (mg/dL)	0.06 [0.03, 0.09]	0.05 [0.02, 0.11]	0.07 [0.04, 0.11]	0.496
WBC (/µL)	5088 ± 1543	5157 ± 1467	5960 ± 2120	0.090
RBC (10^4^/µL)	365.7 ± 33.3	374.1 ± 36.2	371.5 ± 36.6	0.409
Haemoglobin (g/dL)	10.9 ± 0.9	11.0 ± 0.8	10.8 ± 0.9	0.664
MCV (fL)	93.7 ± 6.1	92.1 ± 6.2	90.6 ± 6.5	0.137
Reticulocyte counts (10^4^/µL)	4.0 ± 2.2	4.0 ± 1.6	4.9 ± 2.6	0.181
PLT (10^4^/µL)	14.1 ± 4.0	15.0 ± 4.7	16.5 ± 5.2	0.136
Fe (µg/dL)	54.2 ± 24.7	62.8 ± 23.1	68.3 ± 34.8	0.062
Ferritin (ng/mL)	31.3 [16.3, 58.3]	26.0 [12.5, 41.6]	23.6 [16.3, 82.3]	0.088
TIBC (µg/dL)	255.4 ± 41.9	273.9 ± 44.6	275.8 ± 51.7	0.050
TSAT (%)	21.9 ± 10.6	23.6 ± 9.1	25.3 ± 12.6	0.404
ESA (µg/4 weeks)	137.2 ± 57.6	118.3 ± 58.7	129.5 ± 66.2	0.185
ERI [µg(4 weeks)/kg/(g/dL)]	0.26 ± 0.13 ^c^	0.20 ± 0.11 ^e^	0.21 ± 0.16	**0.016**
ACE inhibitor *n*(%)	4 (8.3%)	11 (11.1%)	31 (15.0%)	0.668
iPTH (pg/mL)	130.9 ± 79.2	131.9 ± 88.6	127.2 ± 68.2	0.974
Zn (µg/dL)	60.8 ± 17.7	65.1 ± 9.8	69.2 ± 16.6	**0.047**
Kt/V	1.48 ± 0.27 ^d^	1.41 ± 0.26 ^d^	1.16 ± 0.28 ^c,e^	**<0.001**
PCR (g/day)	43.6 ± 10.9 ^c,d^	52.0 ± 11.8 ^e^	58.0 ± 22.0 ^e^	**<0.001**
Adiponectin (µg/mL)	25.4 ± 10.7 ^c,d^	20.9 ± 9.5 ^e^	16.9 ± 11.4 ^e^	**0.004**
TC (mg/dL)	159.9 ± 27.8	159.4 ± 34.1	150.6 ± 25.1	0.488
HDL-C (mg/dL)	52.7 ± 12.7	51.4 ± 15.2	49.7 ± 11.7	0.710
LDL-C (mg/dL)	90.1 ± 26.1	89.1 ± 27.4	78.4 ± 21.1	0.214
TG (mg/dL)	82.5 [61.8, 112.8]	80.0 [57.0, 106.0]	97.5 [58.3, 151.5]	0.344
Free carnitine (nmol/mL)	16.87 [14.88, 18.56] ^c,d^	25.62 [22.58, 29.64] ^d,e^	42.77 [39.27, 47.95] ^c,e^	**<0.001**
C2/(C16 + C18:1)	17.8 [15.1, 21.5] ^c,d^	24.3 [19.5, 28.1] ^d,e^	33.7 [30.6, 36.6] ^c,e^	**<0.001**

^a^ Group 1: carnitine deficiency (free carnitine less than 20 nmol/mL), Group 2: carnitine insufficiency (free carnitine of 20 to less than 36 nmol/mL), Group 3: carnitine sufficiency (free carnitine of 36 to 74 nmol/mL). ^b^ All analyses were performed using ANOVA or chi-squared test with Bonferroni’s post hoc analysis as appropriate. Values of HD vintage, CRP, ferritin, TG, free carnitine, and C2/(C16 + C18:1), which were shown as median (IQR), were log10 transformed and used for ANOVA. *p*-values in bold are statistically significant. ^c^ Significantly different from group2, ^d^ significantly different from group3, ^e^ significantly different from group1.

**Table 4 diagnostics-14-02295-t004:** Univariate linear regression analysis of variables affecting reticulocyte counts in the whole group.

Variables ^a^	Univariate Linear Regression Analysis		
	β	95% CI	*p-*Value ^b^
Sex (male = 1, female = 0)	−0.221	−0.371 to −0.071	**0.004**
Age (year)	0.007	−0.147 to 0.161	0.927
log (HD vintage, month)	−0.080	−0.234 to 0.073	0.302
DM (yes = 1, no = 0)	−0.018	−0.173 to 0.136	0.815
HT (yes = 1, no = 0)	−0.060	−0.210 to 0.093	0.445
Smoking (yes = 1, no = 0)	0.127	−0.025 to 0.280	0.102
Dry weight (kg)	−0.072	−0.225 to 0.081	0.356
log (TG, mg/dL)	0.263	0.114 to 0.411	**0.001**
TC (mg/dL)	0.052	−0.101 to 0.206	0.504
HDL-C (mg/dL)	−0.077	−0.231 to 0.077	0.326
LDL-C (mg/dL)	0.001	−0.153 to 0.154	0.994
Adiponectin (µg/mL)	−0.028	−0.181 to 0.126	0.724
Fe (µg/dL)	−0.155	−0.307 to −0.003	**0.045**
TSAT (%)	−0.137	−0.289 to 0.015	0.078
Zn (µg/dL)	0.108	−0.044 to 0.261	0.163
RBC (10^4^/µL)	0.115	−0.038 to 0.268	0.138
Haemoglobin (g/dL)	0.199	0.049 to 0.350	**0.010**
WBC (/µL)	0.173	0.021 to 0.324	**0.026**
PLT (10^4^/µL)	0.239	0.090 to 0.389	**0.002**
log (CRP, mg/dL)	0.028	−0.126 to 0.182	0.719
log (Ferritin, ng/mL)	−0.059	−0.886 to 0.769	0.889
Albumin (g/dL)	−0.015	−0.170 to 0.139	0.847
AST (U/L)	−0.041	−0.195 to 0.112	0.596
ALT (U/L)	−0.083	−0.236 to 0.071	0.288
ALP (U/L)	0.180	0.029 to 0.332	**0.020**
LDH (U/L)	0.128	−0.024 to 0.281	0.099
Kt/V	0.054	−0.098 to 0.207	0.483
PCR (g/day)	−0.065	−0.219 to 0.088	0.402
ESA (μg/4 weeks)	0.084	−0.069 to 0.237	0.280
ERI [μg(4 weeks)/kg/(g/dL)]	0.088	−0.065 to 0.241	0.257
ACE inhibitor (yes = 1, no = 0)	−0.073	−0.227 to 0.080	0.345
iPTH (pg/mL)	−0.094	−0.248 to 0.060	0.229
log (C18:1, nmol/mL)	0.339	0.195 to 0.484	**<0.001**

^a^ *p*-values in bold are statistically significant. Among acylcarnitines showing significant univariate associations with reticulocytes (Table 2), log (C18:1) was selected as a representative factor for multivariate analysis due to regression coefficient magnitude. ^b^ *p*-values in bold are statistically significant.

**Table 5 diagnostics-14-02295-t005:** Stepwise multivariate linear regression analysis of variables affecting reticulocyte counts in the whole group.

Variables ^a^	Stepwise Multivariate Linear Regression Analysis							
	Adjusted R^2^ = 0.291			Bootstrap Results (2000 Replicas)				
	β	95% CI	*p*-Value ^b^	Bias	SE	Median	95% CI	*p*-Value ^b^
Sex (male = 1, female = 0)	−0.146	−0.282 to −0.011	**0.034**	−0.002	0.072	−0.145	−0.292 to −0.015	**0.030**
log (TG, mg/dL)	0.184	0.044 to 0.324	**0.010**	−0.001	0.070	0.180	0.046 to 0.322	**0.009**
Fe (µg/dL)	−0.176	−0.313 to −0.038	**0.012**	0.001	0.067	−0.176	−0.311 to −0.039	**0.011**
Haemoglobin (g/dL)	0.163	0.028 to 0.298	**0.018**	−0.003	0.056	0.160	0.043 to 0.264	**0.017**
WBC (/µL)	-		0.479					
PLT (10^4^/µL)	0.156	0.018 to 0.294	**0.027**	−0.002	0.076	0.153	0.010 to 0.307	**0.026**
ALP (U/L)	-		0.090					
log (C18:1, nmol/mL)	0.286	0.150 to 0.421	**<0.001**	−0.0005	0.070	0.285	0.154 to 0.425	**<0.001**

^a^ Variables showing significant univariate associations with ERI were included in the list. ^b^ *p*-values in bold are statistically significant.

**Table 6 diagnostics-14-02295-t006:** Univariate linear regression analysis to determine independent predictors for ERI in the whole group.

Variables ^a^	Univariate Linear Regression Analysis		
	β	95% CI	*p-*Value ^b^
Sex (male = 1, female = 0)	−0.036	−0.190 to 0.117	0.641
Age (year)	0.112	−0.041 to 0.264	0.151
log (HD vintage, month)	0.166	0.015 to 0.318	**0.032**
DM (yes = 1, no = 0)	−0.199	−0.351 to −0.048	**0.010**
HT (yes = 1, no = 0)	−0.117	−0.264 to 0.036	0.133
Smoking (yes = 1, no = 0)	0.019	−0.135 to 0.173	0.807
log (TG, mg/dL)	−0.206	−0.356 to −0.055	**0.008**
TC (mg/dL)	−0.194	−0.345 to −0.044	**0.012**
HDL-C (mg/dL)	0.027	−0.127 to 0.181	0.730
LDL-C (mg/dL)	−0.148	−0.300 to 0.004	0.056
Adiponectin (µg/mL)	0.163	0.012 to 0.315	**0.035**
Fe (µg/dL)	−0.263	−0.412 to −0.115	**0.001**
TSAT (%)	−0.243	−0.392 to −0.094	**0.002**
Zn (µg/dL)	0.028	−0.125 to 0.182	0.718
RBC (10^4^/µL)	−0.130	−0.283 to 0.022	0.093
Reticulocyte counts (10^4^/µL)	0.088	−0.065 to 0.241	0.257
WBC (/µL)	−0.194	−0.345 to −0.043	**0.012**
PLT (10^4^/µL)	−0.305	−0.451 to −0.159	**<0.001**
log (CRP, mg/dL)	0.090	−0.063 to 0.243	0.248
log (Ferritin, ng/mL)	−0.089	−0.242 to 0.064	0.252
Albumin (g/dL)	−0.038	−0.192 to 0.117	0.628
AST (U/L)	0.128	−0.024 to 0.281	0.099
ALT (U/L)	−0.091	−0.244 to 0.062	0.241
ALP (U/L)	0.075	−0.078 to 0.228	0.336
LDH (U/L)	0.026	−0.128 to 0.179	0.740
Kt/V	0.185	0.032 to 0.337	**0.018**
PCR (g/day)	−0.229	−0.382 to −0.077	**0.003**
ACE inhibitor (yes = 1, no = 0)	−0.028	−0.182 to 0.125	0.718
iPTH (pg/mL)	−0.177	−0.329 to −0.025	**0.023**
log (free carnitine, nmol/mL)	−0.152	−0.304 to −0.0001	**0.050**
log (C2, nmol/mL)	−0.172	−0.323 to −0.020	**0.026**
log (C8:1, nmol/mL)	−0.199	−0.350 to −0.049	**0.010**
log [C2/(C16 + 18:1)]	−0.268	−0.404 to −0.121	**<0.001**

^a^ Dry weight and haemoglobin were used in the formula of ERI and were therefore excluded from the list of variables. Among all types of carnitines, free carnitine, C2, C8:1, and C2/(C16 + 18:1) had significant univariate associations with ERI and were therefore included in the list of variables. ^b^ *p*-values in bold are statistically significant.

**Table 7 diagnostics-14-02295-t007:** Stepwise multivariate linear regression analysis to determine independent predictors for ERI in the whole group.

Variables ^a^	Stepwise Multivariate Linear Regression Analysis							
	Adjusted R^2^ = 0.315			Bootstrap Results (2000 Replicas)				
	β	95% CI	*p*-Value ^b^	Bias	SE	Median	95% CI	*p*-Value ^b^
log (HD vintage, month)	-		0.288					
DM (yes = 1, no = 0)	−0.223	−0.356 to −0.083	**0.002**	−0.0005	0.064	−0.219	−0.345 to −0.095	**0.002**
log (TG, mg/dL)	−		0.743					
TC (mg/dL)	−0.178	−0.177 to −0.041	**0.012**	−0.004	0.069	−0.179	−0.322 to −0.047	**0.010**
Adiponectin (µg/mL)	−		0.172					
Fe (µg/dL)	−0.295	−0.428 to −0.154	**<0.001**	−0.003	0.084	−0.294	−0.452 to −0.131	**<0.001**
WBC (/µL)	−		0.883					
PLT (10^4^/µL)	−0.249	−0.387 to −0.111	**0.001**	−0.0004	0.075	−0.250	−0.389 to −0.101	**0.001**
Kt/V	−		0.751					
PCR (g/day)	-		0.377					
iPTH (pg/mL)	-		0.130					
log(free carnitine, nmol/mL)	-		0.899					
log (C2, nmol/mL)	-		0.265					
log (C8:1, nmol/mL)	-		0.200					
log [C2/(C16 + 18:1)]	−0.219	−0.346 to −0.072	**0.003**	−0.003	0.072	−0.211	−0.358 to −0.078	**0.002**

^a^ Variables showing significant univariate associations with ERI were included in the list. However, TSAT was excluded due to the high collinearity with Fe. ^b^ *p*-values in bold are statistically significant.

## Data Availability

The original contributions presented in the study are included in the article/Appendix A, and further enquiries can be directed to the corresponding author (hkimura@u-fukui.ac.jp).

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
