# Peer review of "Fatty Acid β-Oxidation May Be Associated with the Erythropoietin Resistance Index in Stable Patients Undergoing Haemodialysis"

_diagnostics, 2024, doi:10.3390/diagnostics14202295_

Round 1
Reviewer 1 Report
Comments and Suggestions for Authors
Dear Editor-in-Chief
Thank you for inviting me to review the manuscript entitled " Fatty acid β-oxidation may be associated with the erythropoietin resistance index in stable patients undergoing haemodialysis". In the present study, the authors interestingly provided data on how high total cholesterol and β-oxidation can be related to reticulocyte counts and erythropoietin resistance index. Although this study addressed an interesting concept, some issues should be considered:
1. In the Abstract, the Method section “We investigated the association of reticulocyte counts and ERI with serum levels of lipid markers, adiponectin, and carnitine profiles (C0–C18 measured by LC-MS/MS) in 167 stable patients undergoing HD” is not similar to “Method”. It could be the aim of your study at the end of “Background”. Try to provide a summary of methods.
2. The second paragraph of the Introduction could be divided into two paragraphs.
3. The first paragraph of the Discussion section provided good data on the tested work; nonetheless, it requires a very short introduction as its beginning is somehow confusing and might lead to the distraction of the readers.
4. The reasons for the possible association between the male sex and Fe levels and reticulocyte counts have not been discussed properly in the fourth paragraph of the Discussion (line 346). I suggest rewriting this paragraph.
5. Check the whole manuscript as some punctuation and spacing errors are present.
Comments on the Quality of English LanguageCheck the whole manuscript as some punctuation and spacing errors are present.
Reviewer 2 Report
Comments and Suggestions for Authors
The submitted manuscript by Kidoguchi S and co-authors presented data on the correlation of several parameters with erythropoiesis, including important indicators such as reticulocytes number and erythropoietin resistance index. The work is of great significance in the attempt to identify novel prognostic markers for patients undergoing hemodialysis. The paper is well written and presents a clear argument. The fundamental conclusions are logically supported by the findings of the study. The author was forthcoming about the potential inadequacy of the patient population size, which may have precluded the attainment of a normal distribution in several statistical analyses. The discussion is not merely an account of clinical and biochemical correlations; it illuminates pivotal metabolic mechanisms and their switches occurring at crucial stages of erythropoiesis. The paper will undoubtedly be of interest to readers of Diagnostics.
A few minor comments are offered for consideration:
- The paper will undoubtedly benefit from the inclusion of a graphical abstract, which would present a principal scheme of erythroid differentiation switches in metabolism at certain steps and hormone regulation. This would undoubtedly attract attention and help readers to follow the intricate discussion of the obtained results.
- Line 317 The correct statement should include “proliferation and differentiation of ”
- Several abbreviation decodings are missing:
1) “triglyceride” = (TG) missing in Abstract
Reviewer 3 Report
Comments and Suggestions for Authors
In the study by Kidoguchi and colleagues, using clinical descriptors from patients undergoing hemodialysis, the authors demonstrated a significant association between serum lipid content and erythropoiesis. Their findings show that certain biochemical markers, such as triglycerides and oleic acid, positively influence reticulocyte counts. One of the most noteworthy observations was the critical role of acyl chain length in determining the erythropoietin resistance index (ERI), supporting the conclusion that β-oxidation is essential for the erythropoiesis process. Although the study is clinical, observational, and analytical, it successfully identifies important associations with experimental potential. The work is timely and relevant, aligning with concurrent and independent research in the field, and creates opportunities for experimental studies aimed at improving therapies for patients with chronic renal failure or undergoing hemodialysis
Regarding the structure of the manuscript, it is coherent and clear. While the frequent use of acronyms may occasionally cause confusion, it remains understandable. The introduction is sufficiently clear The findings are primarily presented in well-organized tables, making them easy to interpret, and the conclusions are precise and well-supported by the data.
Reviewer 4 Report
Comments and Suggestions for Authors
All comments and revisions have been provided within the manuscript.

The English quality of the manuscript is acceptable and appropriate.
Round 2
Reviewer 1 Report
Comments and Suggestions for Authors
Dear Editor-in-Chief,
The authors addressed my concerns satisfactorily. The manuscript is now proper to be published in its present form.